# Business Management for Sustainability

John Ikerd 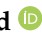

Department of Agricultural and Applied Economics, University of Missouri-Columbia, Columbia, MO 65211, USA; jeikerd@gmail.com

**Abstract:** The purpose of this paper is to address a fundamental flaw in prominent approaches to managing businesses for sustainability. Current management strategies fail to recognize the fundamental differences between economic, social, and moral or ethical values. Economic values are instrumental, individual, and impersonal. Social values are reciprocal, communal, and personal. Moral values are altruistic, spiritual, and universal. These are not arbitrary definitions but expressions of basic differences among the three types of value. These differences reveal the fundamental flaws in attempts to assign economic value or objectively quantify the social and ecological costs and benefits of economic enterprises. The transactional, social, and moral economies are defined in ways that avoid compromising the differences in values. In addition, a natural hierarchy exists among nature, society, and economy that requires a corresponding hierarchy of moral, social, and economic values in managing sustainable organizations. The strategies, motives, and metrics that have dominated sustainable business management for the past several decades, and the related research and educational programs that support them, fail to reflect differences among economic, social, and moral values that are critical to sustainability.

**Keywords:** sustainability; business management; Corporate Social Responsibility (CSR); Impact Investing (II); Triple Bottom Line (TBL); shareholder value; economic value; social value; moral value; ethics

## 1. Introduction

The basic concepts of managing businesses for sustainability are deeply rooted in the history of corporate responsibility. Corporate charters were initially granted to allow investors to pool their capital for a specific public purpose. If a corporation failed to fulfill its public responsibilities, its charter could be revoked [1]. A corporate charter was a privilege granted by society in that corporations limited the financial liability for investors and pooling capital for large investments was otherwise considered economic collusion. The public expected to receive benefits in return for limiting liability and granting the privilege to collude.

The expansions of public responsibility to include social equity and environmental stewardship were logical responses to growing public awareness of the negative impacts of economic exploitation and extraction on society and nature. However, the strategies and metrics for managing and monitoring the performance of businesses have failed to evolve to accommodate today's social and ecological mandates for public responsibility.

The first section of this paper provides an overview of some of the most prominent strategies for managing businesses for sustainability. The next section frames sustainable management within the broader context of corporate responsibility, with a specific emphasis on the evolution of corporate responsibility in the United States of America (USA). The overview of the current state of sustainable business management is concluded with a discussion of some of the more popular metrics or measures of social and ecological performance. This paper is not intended to be a comprehensive review of the literature on these subjects [2]. Nor is it intended to be a research paper that quantifies deficiencies

in the performance of sustainably managed businesses [3]. It is a perspective paper intended to provide potentially valuable insights into how to manage the multidimensions of sustainability more effectively.

The sections of the paper that deal directly with management strategies for sustainability include the different motives for management, the three economies in which sustainable businesses function, the limitations of economic metrics of sustainability, and the implications for sustainable business management. The conclusions section of the paper addresses the priorities among nature, society, and economy and the associated hierarchy of moral, social, and economic values essential for sustainability. These hierarchies reveal the essential role of government in maintaining an economic environment in which businesses can be managed for sustainability.

## 2. Management Strategies for Sustainability

In 1987, The United Nations Brundtland Commission defined sustainability as "meeting the needs of the present without compromising the ability of future generations to meet their own needs [4]". Everything capable of meeting human needs, including everything of economic value, is ultimately derived from the resources of the Earth—sunlight, air, water, minerals, and biological ecosystems and organisms. Society provides the human resources—workers, managers, investors, citizens—necessary to transform natural resources into forms that are useful to humans, including goods and services that have economic value. Thus, the logical rationale for the ecological, social, and economic pillars of sustainability. Merriam-Webster defines sustainability as "relating to, or being a method of harvesting or using a resource so that the resource is not depleted or permanently damaged [5]". Thus, a sustainable business, economy, or society must contribute to meeting the needs of the present without depleting or permanently damaging the productivity of the natural and human resources that will be needed to meet the needs of the future.

Corporate Social Responsibility (CSR) is a logical business response to the ecological and social challenges of managing businesses for sustainability. A CSR business has social and ethical responsibilities. "The more visible and successful a corporation is, the more responsibility it has to set standards of ethical behavior for its peers, competition, and industry [6]". Social responsibilities include fair treatment of all customers and vendors regardless of age, race, culture, or sexual orientation, equitable pay and benefits for employees, and transparency for investors. Ethical responsibilities include conserving natural resources, reducing pollution, recycling materials, and reusing products. Philanthropic responsibilities include donations of profits to charities, support of employee philanthropy, and charitable fundraising. In addition to remaining financially sound, financial responsibilities include investments in research and development for products and processes that promote ecological and social sustainability.

Socially Responsible Business (SRB) is similar to CSR in that the two share a common goal to make positive contributions to society, minimize the harmful effects of economic activities on the environment, and bring about positive changes in industry as well as society [7]. The main difference is that the SRBs form partnerships and alliances with local communities and collaborate with local non-governmental organizations (NGOs) and local government agencies. SRBs prioritize community connectedness and develop long-term relationships with other organizational members of their communities. SRBs are more socially and personally connected to their communities than CSR businesses.

Tri-sector partnerships (TSPs) are like SRBs except the focus of TSPs is on sustainable development rather than sustainable business management. TSPs normally include for-profit companies, government agencies, and local civic or charitable organizations. The commitments of TSP businesses to social and ecological sustainability depend on the sustainability commitments of the partnerships they join or help form. TSPs emerged in the international development community to address the economic exploitation of natural and human resources of host nations by global corporations [8]. However, the TSP concept is equally relevant to domestic community development.

Benefit corporations (BCs) provide a legal structure that allows corporations to "opt out of shareholder primacy and opt into stakeholder governance" [9]. Managers of BCs are required to consider the consequences of their decisions for their workers, customers, communities, societies, and the natural environment. The specific social and environmental goals and objectives of BCs are developed in consultation with boards of directors, who represent the interests of the various types of stakeholders. Benefit corporations, like other businesses managed for sustainability, must commit to higher standards of purpose, accountability, and transparency than other for-profit corporations.

Sustainably managed businesses are responding to growing public demands for social and ecological responsibility. A national industry survey of 1200 consumers, 100 corporate executives, and 100 professional investors concerning public attitudes toward corporate responsibility found that 75% of consumers responded negatively to corporations they feel are not socially responsible [10]. In addition, 39% indicated they would not buy from such firms and 25% indicated they would tell their friends and family to avoid such companies. In total, 81% of employees believe their employers are socially responsible but only 40% believe all companies in the USA are socially responsible. More than 80% percent of professional investors indicated a preference for owning stock in corporations that are known for social responsibility.

Investments in sustainable business management are referred to as impact investing (II) or socially responsible investing (SRI) [11]. The term SRI generally refers to avoiding investments that conflict with investors' ethical beliefs, whereas II refers to investing in businesses that generate social and environmental benefits rather than avoiding investments in businesses believed to be harmful to nature or society.

Investment experience, as well as motives, are important determinants of both impact and socially responsible investors' decisions [12]. Both are motivated by expectations of positive economic returns as well as social and ecological returns on their investments. The experience of economic losses by CSR, SRP, or BCs can affect future investments and businesses may be tempted to prioritize economic performance over environmental and social benefits. However, a lack of transparency may result in greater defections of socially responsible and impact investors than periodic disappointments in financial performance.

## 3. Evolution of Corporate Responsibility

Despite the growing popularity of various approaches to managing businesses sustainably, maximizing returns for shareholders has dominated business management strategies for the past several decades, particularly in the USA. The various concepts of corporate social and ethical responsibility met with strong resistance from prominent economists during the 1970s. "Social responsibility is a fundamentally subversive doctrine in a free society", according to Milton Friedman, a highly respected economist and influential defender of modern free-market capitalism [13]. Friedman was awarded the 1976 Nobel Memorial Prize for Economic Science. "There is one and only one social responsibility of business", he wrote in a 1970 article in the New York Times, "to use its resources and engage in activities designed to increase its profits so long as it stays within the rules of the game, which is to say, engages in open and free competition without deception or fraud [13]". Friedman criticized businessmen who were promoting the "social responsibilities of business in a free-enterprise system [13]". He criticized businesses that claim to accept social responsibilities for providing employment, eliminating discrimination, and avoiding pollution for claiming they were defending free enterprise. "In fact", he wrote, "they are—or would be if they or anyone else took them seriously—preaching pure and unadulterated socialism [13]".

Friedman's comments were a response to the corporate social responsibility movement that emerged during the 1950s and grew with the environmental movement of the 1960s [14]. However, the responsibilities of businesses to their workers and customers, as well as their investors, are rooted even deeper in the history of economics and business management. Adam Smith, for example, suggested that businesses needed to support education programs

for their employees to offset the mind-numbing effects of the prescribed, repetitive work in factories in his 1776 book, Wealth of Nations [15]. In Theory of Moral Sentiments [16], Adam Smith wrote: "The wise and virtuous man is at all times willing that his own private interest should be sacrificed to the public interest of his own particular order or society [16] (p. 406)".

Frederick W. Taylor, a business luminary of the early 1900s, wrote, "The best management is a true science, resting upon clearly defined laws, rules, and principles, as a foundation [17]". He pointed out that business success is almost entirely dependent on "getting the initiative of the workmen", which he suggested is rarely, if ever, attained. The key elements of his approach to management were teaching, training, and establishing a shared understanding of production methods by workers and managers. Taylor's writings suggest that his scientific approach to management was meant to allow workers to participate in management and to benefit from improved working conditions and the resulting increases in efficiency and productivity. However, Taylor's focus on efficiency has instead been misused by managers to pressure workers to perform repetitive tasks more quickly and efficiently without participating in management or sharing the benefits.

Total Quality Management (TQM), championed by W. Edwards Deming and others in the 1950s, focuses on businesses' responsibilities to employees and customers. "Employees must buy into the processes and system if TQM is going to be successful. A company adopting TQM principles must be willing to train employees and give them sufficient resources to complete tasks successfully and on time [18]". "Under TQM, your customers define whether your products are high quality [18]". Customer input is relied on to continually improve raw materials, manufacturing processes, and quality control procedures and thus customers are treated as TQM stakeholders.

Peter Drucker was a management consultant, educator, and author who provided much of the philosophical and practical foundation of modern management theory [19]. He wrote in his 1973 book, Management: Tasks, Responsibilities, Practices [20], "that in modern society there is no other leadership group but managers. If the managers of our major institutions, and especially of business, do not take responsibility for the common good, no one else can or will [20] (p. 325)".

Drucker taught managers to go beyond scientific management by focusing on opportunities rather than problems and putting themselves in the place of their customers, workers, and suppliers as a means of understanding and continually refining their competitive advantages [21]. Drucker taught the principles and practices of what today would be called "stakeholder capitalism" [22]. Stakeholder capitalism recognizes that businesses have responsibilities to everyone who contributes to their success.

However, many prominent economists and business managers never bought into the idea that for-profit corporations had responsibilities to anyone other than their shareholders. In addition, Milton Friedman's critique of socially responsible business provided a highly credible, academic defense for corporate managers who managed to maximize the single economic bottom line.

After a two-year consultancy with General Motors, Drucker wrote a book suggesting that GM should re-examine its long-standing policies on customer relations, dealer relations, and employee relations. GM's chairman, Alfred Sloan, was so upset about the book he "simply treated it as if it did not exist [23]". Jack Welch, CEO of General Electric Corporation, was one of the most prominent advocates of Friedman's philosophy of business management during the 1980s and 1990s [24]. Through cost-cutting, mergers, and acquisitions, Welch increased the market value of General Electric from $14 billion to $410 billion during his 20-year reign and turned GE into the world's second-largest company.

Welch later called it a "dumb idea", but Welch was regarded as the father of the "shareholder value movement" that dominated business management for more than 20 years [25]. Many if not most corporate managers were then, and still are, evaluated primarily, if not solely, by quarter-to-quarter changes in the value of their corporations' stock.

The modern concept of corporate responsibility in the USA dates to a 1953 book written by Howard Bowen, an economist and president of Grinnell College, Social Responsibilities of the Businessman [26]. Bowen referred to the writing of Adam Smith in justifying his advocacy for business ethics and responsibilities to societal stakeholders in addition to workers, suppliers, and customers. However, the idea that corporations had responsibilities to anyone other than their shareholders was difficult for corporate managers to resist during the 1980s and 1990s. Corporate Social Responsibility resurfaced as a business management strategy with the emergence of sustainability as a significant public concern during the 1990s. It wasn't until the sustainable development movement of the late 1990s that challengers of the shareholder value movement began to gain significant credibility in the corporate community.

## 4. Business Metrics for Sustainability

Like other businesses, businesses that are managed by the principles of CRS, SRB, TSP, BCs, or other approaches to managing for sustainability need ecological, social, and economic indicators or metrics to guide their decisions, document their progress, and attract impact and socially responsible investors. Environmental, social, and governance (ESG) refers to a set of criteria used to analyze the sustainability of a company's behaviors and policies as well as its productivity [26]. ESG criteria are designed to verify that the companies are not only responsible stewards of the environment and good corporate citizens but are also led by corporate managers who use transparent accounting systems.

The governance metrics of ESG businesses may include accurate and transparent accounting methods, integrity, and diversity in selecting corporate leadership, and accountability to shareholders. The environmental metrics may include corporate climate policies, energy use, waste, pollution, natural resource conservation, and treatment of animals. Social metrics include workplace conditions regarding employees' health and safety and whether the company takes unethical advantage of its customers.

The Triple Bottom Line (TBL) is another approach to evaluating sustainably [27]. The TBL includes the traditional measures of economic value, including profits, return on investment, and shareholder value, but also includes measures of environmental and social values. TBL businesses attempt to measure their performance in terms of profits, people, and the planet—the 3Ps. Some organizations attempt to monetize all three dimensions of the TBL, including social and environmental impacts. Others challenge the logic of trying to place an economic value on nature or society but thus far have not explicitly recognized the fundamental differences between economic, social, and moral values.

Several international standards are available for managers to monitor the performance of their businesses for ecological, social, and economic sustainability. The ISO-International Organization for Standardization provides a set of metrics to "guide businesses in adopting sustainable and ethical practices, helping to create a future where your purchases not only perform excellently but also safeguard our planet [28]". SA8000 standards developed by Socially Accountability International "provide a framework for organizations of all types, in any industry, and in any country to conduct business in a way that is fair and decent for workers and to demonstrate their adherence to the highest social standards [29]". The Global Reporting Initiative provides "sustainability reporting standards, which cover topics that range from biodiversity to tax, waste to emissions, diversity and equality to health and safety [30]". International standards are used primarily to provide managers with transparency in communicating with their internal and external stakeholders, including workers, suppliers, customers, and investors. Internal and international standards may also be used by managers to link their corporate identity with sustainability through corporate advertising and other public relations programs.

No single business that functions within an unsustainable society is sustainable, but businesses that are managed for sustainability can contribute to the societies of which they are a part. For brevity, businesses that adopt the principles and practices of CSR, SRB, TSP, or other strategies for ecological, social, and economic sustainability may be labeled

as sustainably managed businesses or SMBs. Their managers may be called sustainable business managers (SBMs). Regardless of what they are called, businesses that are managed for sustainability and their investors face a common challenge of continually examining their motives and monitoring the multiple ecological, social, and economic outcomes of their management and investment decisions.

## 5. Motives for Sustainable Business Management

In matters regarding sustainability, motives matter as much or more than outcomes. Those who argue that governments need to create economic incentives for ecologically and socially responsible decisions fail to understand that decisions motivated by economics are the primary cause of ecological and societal degradation. Philosophers have long debated whether motives are more or less important than the consequences or outcomes. Some philosophers, including Aristotle and Kant, concluded, or at least suggested, that motives do not matter, or at least do not matter as much as the consequences [31]. Others have argued that motives have importance apart from their consequences or that motives are inseparable from consequences.

Noted anthropologist, Gregory Bateson, contends that the purpose of actions is not only to bring about specific outcomes but also to spread or promote the ideas that motivate the actions [32]. For example, the basic purpose of economic development is not simply to complete specific projects but to spread the ideas that motivate economic development in general. Likewise, the motives of SMBs can affect the motives of other businesses, regardless of the direct or immediate outcomes. The motives of one generation of SBMs can motivate the decisions and actions of future generations of businesses.

The resistance to sustainability by current business managers is not motivated solely by the possibility or likelihood of negative economic outcomes but by a dominant business culture that prioritizes profit maximization over social and ecological integrity. Government programs that provide economic incentives for ecologically and socially responsible business decisions, while intended to bring about positive outcomes, tend to perpetuate a business culture that prioritizes profits over people and planet.

The current culture of economic prioritization is unwittingly reinforced by businesses that limit environmentally and socially responsible decisions to situations that also have the potential to increase profitability. If an alternative action promises a greater economic return, they take it, even if it promises less positive ecological or social consequences. On the other hand, businesses that prioritize social and ethical benefits over maximizing economic returns help create a business culture that prioritizes ecological and social sustainability. If the outcomes of specific actions do not meet their economic expectations, SBMs do not abandon their social or ethical priorities. They keep trying to find economically viable means of expressing the social and ethical values of sustainability through their decisions. Motives matter, whether ecological, social, or economic.

## 6. Three Economies of Sustainability

Management decisions that are critical to sustainability have three basic characteristics. The first is motive; what is the primary motivation or reason for the decision? The second is relationship; what is the nature of the relationship with the person, place, or thing affected by the decision? The third is the span of the impact or outcome of the decision; how widely will the resulting benefits or costs be shared?

### 6.1. Motives for Management Decisions

The different motives for individual and organizational decisions are illustrated in Figure 1. The layers of the cake in this illustration reflect only the motivations for management decisions. The top layer and center core of the cake represent the three basic kinds of economic motives for individual and organizational decisions, including business decisions. The top layer represents decisions that are motivated solely by economics, e.g., the economic bottom-line business. Economic decisions invariably affect society and nature,

as well as the economic bottom line. However, the motivations for decisions matter most in managing sustainability, regardless of the outcomes.

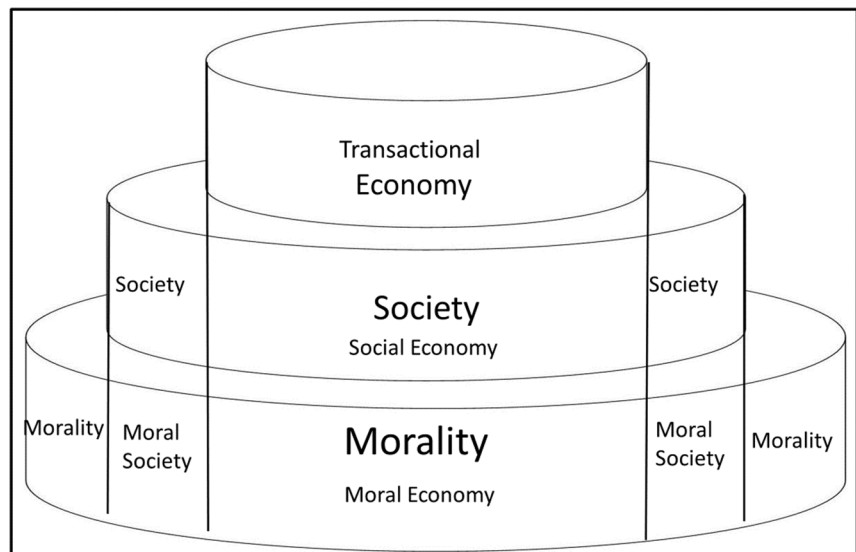

**Figure 1.** Three Economies of Sustainability.

The second layer in Figure 1 represents decisions motivated primarily or solely by societal interests, such as the well-being of families, communities, and nations. The second layer includes decisions motivated solely by societal interests, which is the outside ring of the second layer, as well as decisions motivated by both economic and societal interests, which is the inner ring of the second layer. The inner core of the second layer is referred to as the social economy and is the realm of social responsibility for sustainably managed businesses.

The bottom layer of the cake represents decisions that are motivated by morality or a sense of ethical responsibility for stewardship, guardianship, or caretaking of society and nature. The outer ring of the bottom layer represents decisions that are motivated solely by a sense of moral responsibility, such as those of religious or philanthropic organizations. The inner core of the bottom layer represents decisions that are motivated by both economics and morality. This layer is referred to as the moral economy, where economic decisions are tempered by a sense of ethical or moral responsibility. The motivations of businesses managed for sustainability are depicted as the inner core of the bottom two layers of the cake in Figure 1—the social and moral economies.

The middle ring of the bottom layer, the moral society, represents the decisions of individuals and organizations that are motivated by a sense of morality and social responsibility. This is the realm of public administration or government. In matters of government, elected representatives and administrators should strive to make decisions that reflect their ethical or moral sense of what their government should do to benefit society as a whole, including those who may or may not share their moral values. Individuals or businesses that use government to serve their economic interests, or to impose their moral values, or lack thereof, on others, are motivated neither by morality nor social responsibility but by narrow individual self-interests.

*6.2. Characteristics of Economic, Social, and Moral Values*

In managing businesses for sustainability, managers must respect the fundamental differences between economic, social, and moral values. The term "value" has a variety of meanings and connotations, but as used here, is a measure of "relative worth, utility, or importance [33]". There are also many different definitions and dimensions of economic, social, and moral or ethical values, but the characteristics explained here are of specific relevance to sustainability. Once SBMs understand the differences, they will be better able

to meet the challenge of developing appropriate management strategies and metrics for monitoring their progress toward sustainability.

The basic characteristics of values associated with economically, socially, and morally motivated decisions are shown in Table 1. Economic values are individual, instrumental, and impersonal—the center column of Table 1 [34]. Economic relationships are instrumental in that they are means to ends rather than the goals or ultimate objectives to be achieved. For example, profits provide money for hiring workers and acquiring new technologies as a means of increasing productivity. Economic investments are made with the expectation of receiving something of greater economic value in return. However, when returns on economic investment are monetary, the returns have no value other than the value of whatever money can be used to acquire, either immediately or at some time in the future.

**Table 1.** Economic, Social, and Ethical Decisions.

| Decision Characteristics | Social Values | Social Economy | Economic Values | Moral Economy | Moral Values |
|---|---|---|---|---|---|
| Motivations | Reciprocal | Inst/Reciprocal | Instrumental | Inst/Altruist | Altruistic |
| Relationships | Personal | Imp/Personal | Impersonal | Imp/Spiritual | Spiritual |
| Outcomes | Communal | Ind/Common | Individual | Ind/Universe | Universal |

Economic value is impersonal in that economic relationships or transactions are not dependent on the specific persons, businesses, or "others" involved. To the extent that a relationship is personal, it is non-economic because it is not motivated solely by self-interests. Noted economist Philip Wicksteed wrote in 1933, "As soon as he is moved by a direct and disinterested desire to further the purposes or consult the interests of particular 'others'... the transaction on his side ceases to be purely economic" [35] (pp. 173–174). People who work for businesses may have personal ties that result in economic benefits for themselves and their employers, but the personal value of their relationships is non-economic. Businesses that are managed for the economic bottom line buy from suppliers that offer the lowest prices, pay their employees only as much as necessary, and sell to customers that will pay the highest prices for what they have to offer, without regard to the specific persons involved in the various economic relationships and transactions.

Finally, economic value is individual in that economic benefits accrue to the individual enterprise or business, not to families, friendships, communities, or societies in general. An aggregate economy is nothing more than a collection of individual economic enterprises and organizations. Positive relationships among workers, suppliers, and customers of businesses may create social and moral values that contribute to the sustainability of the community and society. However, relationships have no economic value other than what they contribute to the economic bottom line of the business. Relationships among individuals, businesses enterprises, or organizations that make up an economy have no economic value beyond whatever they may contribute to the economic efficiency of business relationships and transactions of individual businesses. A strong economy is important to profit-maximizing businesses only to the extent that the economy contributes to its economic bottom line.

Social values differ from economic values in that social relationships are reciprocal, rather than instrumental, personal rather than impersonal, and the outcomes are shared or communal rather than individual [34]. Social relations are reciprocal because there is an expectation of receiving something of personal value from social relationships. Unlike economic contracts of legal obligations, however, there are no specific expectations regarding when, how, or how much social value a friendship, membership, or citizenship will yield in return. Anything of economic value associated with a social relationship is part of the social economy. However, the social relationship will have no economic value because social returns are specific to the persons or individuals involved.

Unlike economic relationships, social relationships are personal rather than impersonal. Social values arise from friendships, families, communities, and societies in which there

is a personal sense of belonging. Since personal relationships exist only between specific individuals, social values are not transferrable to different individuals. A person cannot buy or sell a friendship or sense of belonging. Since social relationships cannot be bought, sold, or traded, they have no economic value. Within friendships and families, the individuals typically know each other intimately and their social relationships have a major influence on the quality of life. Close personal relationships among employees of businesses and between employees and suppliers or customers may be very important to their quality of life. However, such relationships have no economic value because they cannot be bought, sold, or exchanged.

Close personal relations, such as those in friendships and families, are characterized by sociologists as thick-trust or thick-reciprocity [36]. Less personal relations, such as those who identify personally with communities, societies, or nations are referred to as thin-trust or thin-reciprocity. Regardless of whether thick or thin, if a relationship is social, rather than economic or ethical, there is a sense of personal connectedness or belonging. If there is no sense of personal connectedness among family members, employees, or community members, there is no social family, collegiality, or community but instead merely collections of individuals.

Within communities and societies, the individuals may be only casually acquainted or perhaps just share some common attitudes or aspirations, but their sense of belonging may nonetheless be significant. For example, socially responsible taxpayers willingly pay taxes with no specific expectation of what they will receive in return other than the expectation that if they contribute to society, then society, through government, will reciprocate in some way at some time in the future. Socially responsible businesses pay, rather than attempt to avoid, paying taxes for similar reasons. They also respect laws and regulations rather than attempt to circumvent government oversight or restraint. They understand that businesses benefit in many ways from a socially responsible government. Socially responsible businesses also make investments in their communities and society that are not required by law but are made with the expectation of reciprocal social benefits from their community and citizen stakeholders.

Finally, social values are communal rather than individual because they accrue to friendships, families, collegialities, or communities as wholes rather than to the individuals involved in the relationships or transactions. A family is an emergent property that arises from the relationships among its members. If there are no meaningful relationships, there is no family—only a collection of individuals born or raised in the same household. Whenever the relationships change, the family is changed. Likewise, a community, professional organization, or corporate collegiality is an emergent property of the relationships among unrelated individuals who share a sense of fellowship or belonging. The essence of the whole emerges from shared attitudes, interests, obligations, and aspirations. Those who do not share a sense of belonging are not part of the social group, regardless of where they live, meet, or work.

Moral values are different from economic and social values in that they are universal, rather than individual or communal; altruistic, rather than instrumental or reciprocal; and spiritual, rather than impersonal or communal [34]. Altruism refers to selfless acts that benefit others with no expectation of receiving anything of economic or social value in return. Moral acts are altruistic because they are guided solely by what a person believes is right or good. There is no expectation of receiving anything for themselves or anyone they know, other than the sense that they have done something they should do. Altruism is not limited to religious or charitable contributions of time or money but includes anything that is done without expectations of receiving anything other than a sense of satisfaction in return. Religions and charitable contributions often are not purely altruistic but instead means of acquiring something of economic or social value and thus part of the moral economy or moral society. The moral values of such relationships have no economic or social value because they are neither instrumental nor personal.

Moral values are universal rather than individual or communal because acts of altruism contribute not only to the well-being of communities and societies but also to the betterment of humanity and the whole of the universe. What is right and good for people a person knows is right for those they do not know, including those of future generations, and for the whole of the Earth. Moral relationships are spiritual or psychological because a person's perceptions of what is right and good ultimately come from their sense of connectedness with some higher order of things or their sense of absolute reality or God.

Moral relationships are not limited to relationships between specific persons or persons in general but also include relationships between people with their natural environments. Relationships with the other living and nonliving things of nature—the whole of the Earth, the Universe, and beyond—are moral. Thus, the extraction of natural resources to meet human needs and the disposal of biological and chemical waste into the natural environment have moral consequences, regardless of the motivation. Moral responsibilities to conserve natural resources and protect the natural environment are not solely responsibilities to help secure the wellbeing of other people of current and future generations but are also responsibilities of humans as caretakers of the other living and non-living things of the Earth.

### 6.3. Sustainability Is Ultimately a Matter of Morality

These distinctions among economic, social, and moral values are critical to sustainability because sustainability is ultimately a question of morality. There is no economic rationality in doing anything for the sole benefit of anyone else, including anyone of some distant future generation. Economic value is individual and instrumental and the individuals who make economic investments today will not be present in future generations to realize the economic returns. Thus, there is no logical reason for businesses to make economic investments if the economic returns will be realized by people of some unknown future generation.

There is no social value in doing anything for future generations because there is no possibility of people receiving a personal benefit from someone they can never meet or know. People may receive social benefits associated with investments from which their children will receive economic benefits, but beyond the generations with whom they have social relationships, there is no possibility of benefitting socially from decisions to ensure the wellbeing of future generations. However, people, including business managers and their workers, suppliers, customers, and investors are thoughtful, caring human beings. They need to feel that what they are doing with their time and money is right and good, not just economically or socially expedient.

The hierarchy or priority of values essential for sustainability is shown in Figure 2. Moral values must take priority over social values and social values must take priority over economic values. A diversity of social values may exist among sustainable societies, communities, and organizations but they must share a moral or ethical commitment to meeting the needs of the present without compromising opportunities for the future. A diversity of economies and economic relationships may exist within sustainable societies, communities, and organizations but they must complement and support the social and moral commitment to the sustainability of societies, communities, and organizations, including businesses, of which they are a part.

Economic investments in sustainability may yield positive economic returns in the long run, but the economy places a premium on the present relative to the future. The future is inherently uncertain, and the more distant the expected future economic return, the less its present economic value. Lenders require interest payments in addition to principal repayment because of the risks of default. Borrowers must pay interest to gain access to capital because of uncertainty regarding their ability to repay. Planning horizons of publicly traded corporations average only five to six years because any return further in the future has far less economic value than a return next quarter or next year [37]. Investments that will only benefit future generations are morally imperative for sustainability but

are economically worthless. Businesses that are managed for sustainability cannot be motivated solely by economic, social, or moral values but must consider all three. They must meet the economic and social needs of people of current generations without depleting or permanently damaging the natural resources upon which the economic and social wellbeing of future generations depend.

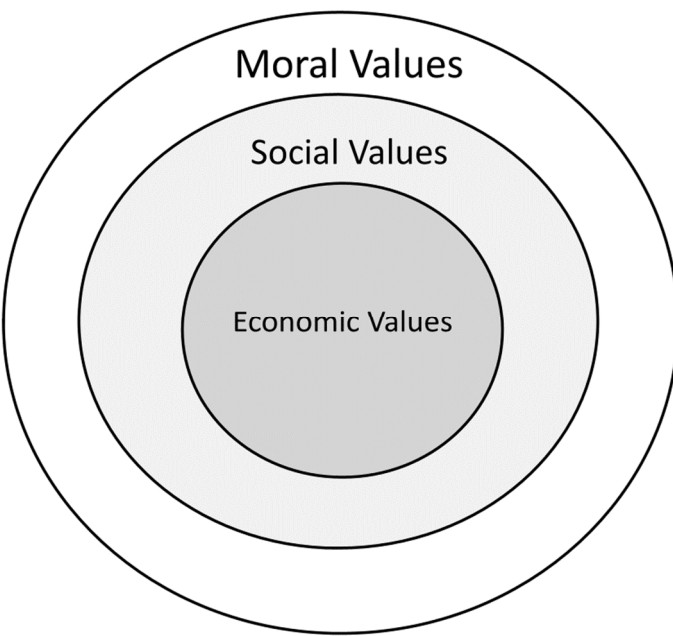

**Figure 2.** Hierarchy of Sustainable Values.

*6.4. The Social and Moral Economies*

Decisions that are motivated by the potential for both social and economic benefits are labeled as the social economy in Table 1. Benefits in the social economy accrue both to individuals and communities, as in the case of employment that provides individual livelihoods and supports the economies of communities. Motivations in the social economy are both instrumental and reciprocal, in that there are expectations of instrumental economic returns as well as reciprocal social benefits from the transactions and relationships. Suppliers and customers may prefer to do business with people they know personally and trust, which results in both social and economic values. While the economic expectations are impersonal and have economic value, the social expectations are personal and do not.

Decisions that are motivated by the potential for both moral and economic benefits are labeled as the moral economy in Table 1. Businesses as well as individuals may benefit economically from actions that reflect their moral values. Relationships and transactions in the moral economy are both impersonal and spiritual, in that decisions motivated primarily by ethics or morality may be rewarded both economically and spiritually. Many consumers and investors are willing to pay more for products and accept less from their investments in companies that reflect their moral values. The economic benefits of management decisions in the moral economy are realized by the businesses but the spiritual and ethical benefits accrue to the managers, employees, communities, societies, and beyond to future generations.

Decisions that are motivated by both social and ethical values are not included in Table 1 but would be labeled as the moral society. To the extent that personal relationships reflect the moral values of the individuals involved, the benefits will be both social and moral. In cases where individuals compromise their moral values to accommodate their social preferences, the social benefits may be offset by the moral costs. The motivation for relationships in the moral society is both altruistic and reciprocal. There may be an expectation of receiving social value from the relationship, but the relationship must not be motivated solely by reciprocity. Such relationships have no economic value but may

be of critical importance to the quality of life of those involved. Moral values provide the conceptual foundation for sustainable communities, societies, and effective governance.

## 7. Inadequacies of Economic Strategies for Sustainability

Managers who have learned the art and science of managing for the economic bottom line or shareholder value may have difficulty adjusting to management strategies for sustainability. Arguably, the two most important quotes in business management are attributed to Peter Drucker, "You can't manage, what you can't measure" and "Management is doing things right, and leadership is doing the right things [38]". In managing sustainable businesses, leadership takes priority over measurement. Managers must first make sure they are measuring the right things and then find means of measuring them rather than finding things they can measure and then managing them. Social and moral values require metrics quite different from measures of economic performance.

Perhaps the greatest challenge to conventional thinking is that social and moral values cannot be translated into economic costs or benefits and thus cannot be integrated to form composite indicators of sustainability. *Natural Capitalism*, a best-seller book in the early 2000s, provided estimates of the economic value of global natural resources that exceeded the gross world product or value of global economic output at the time [39]. Numerous comprehensive studies have provided estimates of the "true costs" of food, which include the economic values of the ecological and social costs that are not paid by food producers and thus not reflected in retail food markets. A report commissioned by the Food and Agricultural Organization of the United Nations placed the non-market cost of food at USD 10 trillion—about 10% of global GDP [40]. However, such estimates only provide estimates of the non-market economic costs and fail to account for the non-economic social and moral costs that are imposed on society and nature.

Economists refer to efforts to assign economic values to unpaid or non-market social and ecological costs and benefits and to reflect them in markets as internalizing externalities. The ultimate objective is to force businesses to pay for the economic costs they currently impose on society and the environment to allow businesses to benefit economically from their economic contributions to society and nature. If these external costs were internalized, the costs would be reflected in the prices that businesses pay for their production inputs. If external benefits were internalized, businesses would be rewarded economically for their positive contributions to nature and society.

A variety of public policies have been used or proposed to internalize the economic externalities of unsustainable businesses. "Contingent valuation" or "willingness to pay" is one approach economists use to assign economic values of resource conservation, environmental protection, and other nonmarket or external benefits of ecologically and socially responsible actions [41]. While contingent valuation may provide useful estimates of the economic costs and benefits of actions that also have social or moral costs and benefits, contingent valuation does not reflect the social or moral costs or benefits.

For example, if someone is asked how much money they would demand to end a friendship, the answer would provide an assessment of the economic value of the friendship but not the social value. It would also provide some indication of how much money the individual thinks it would take to offset one less friend without diminishing their quality of life. However, there is no way to determine the market or exchange value of the relationship because it cannot be sold to anyone else. Friendships that can be bought or sold are economic relationships rather than social relationships.

An economist can ask a person what they would pay to preserve old-growth forests that they never expect to see. But they cannot sell or trade their unique sense of moral responsibility as a caretaker of nature to anyone else, and thus it has no economic value. Access to nature has economic value that can be reflected in access fees, but stewardship of nature is a moral responsibility that has no economic value. Furthermore, moral decisions affect the whole of society, including future generations, and cannot be assessed by summing individual assessments.

Even if monetary values could be placed on social and ethical values, those with more money quite logically would be willing to pay more than those with less money can afford to pay for the same environmental and social benefits. It is illogical to suggest that the willingness to pay more means environmental and social responsibilities are of greater moral and social value to wealthy people than to poor people. People with more money have more influence on market economies because market values are determined by scarcity, not by social equity or moral justice. Those who have more can get more. Social and moral values are not dependent on how much a person is willing and able to pay but on how much a person cares.

Attempts to quantify the economic value of nature and society may be well-meaning and the results may be useful in organizational management as well as the development of public policy. However, the unearned economic benefits realized from ecologically and socially responsible business decisions may be less important than the social and moral benefits derived from the same decisions. Likewise, the unpaid economic costs imposed on nature and society by socially and ecologically irresponsible decisions may be far less important than the associated social and moral costs. Simply "getting the market prices right" or internalizing the economic value of social and ecological externalities, while beneficial, would not ensure that businesses are managed for sustainability [42].

## 8. Implications for Sustainable Business Management

Problems of sustainability are complex but are often treated as if they were complicated rather than complex [43]. "Complicated problems can be hard to solve, but they are addressable with rules and recipes. They also can be resolved with systems and processes" [44]. Ecosystems, societies, and economies are complex rather than complicated. The interrelationships among the economies, societies, and natural ecosystems add to the complexity of sustainability. Complex problems, such as sustainability, involve too many interdependent factors and relationships to be reduced to rules, procedures, or practices. Complex systems also have holistic properties that emerge from relationships that are not present in the parts. Whenever the relationships change, the essence of the whole is changed. Sustainability is a direction rather than a destination, a purpose rather than a process, and a question that has no answer. There is no way of proving what is or is not sustainable because the ultimate outcomes of today's decision can only be known as sometime in the distant future.

There is no set of rules, procedures, or best management practices that can ensure the sustainability of a business organization. There are no universal ecological, social, or economic indicators that can be used to assess the relative sustainability of different businesses or other organizations. Any set of social or ecological indicators that is appropriate for one business, community, or society at one point in time may or may not be relevant for another business, community, or society at a different point in time.

Sustainably managed businesses function in the social and moral economies as depicted in Figure 1. They can use the metrics of economics to measure the economic consequences of their decisions for society and nature, but not the social or moral consequences of their decisions. Triple bottom-line businesses can avoid the social and ecological costs imposed by economic bottom-line businesses but cannot measure the social and moral benefits of their actions using economic metrics. Businesses that exploit society and extract from nature to maximize profits impose economic, social, and ecological costs on society. But forcing extractors and exploiters to pay the economic costs they impose on society, through taxes, licenses, user fees, and such, does not compensate for their social and ecological impacts on society and nature.

The metrics for social responsibility and moral integrity must reflect the characteristics of social and moral values, which require fundamentally different approaches to quantification. A sense of personal connectedness must be established and sustained among employees of SMBs and between their employees and suppliers, customers, investors, and the communities of which they are a part. The social reciprocity associated with these

relationships need not be the thick reciprocity of families or close friendships, but personal connectedness must be sufficient to establish a sense of shared responsibility for the sustainability of the business. The benefits of social responsibility must be communal, meaning the benefits must accrue not only to the business but also to all its stakeholders, including those of future generations.

Different people have different social value systems that define what is deemed appropriate and inappropriate in various kinds of social relationships. Therefore, individual SMBs will need to collaborate with their stakeholders in developing their own unique sets of social indicators of progress toward sustainability. Industry-wide comparisons and rankings among similar businesses may provide useful information but may or may not reflect the social sustainability status of a specific business [45]. Numerous sets of indicators of social sustainability have been developed and published for consideration by managers of SMBs [46]. Such indicators may prove useful in developing consensus among stakeholders as to how best to assess and monitor the social sustainability of the business. However, social indicators used by a business must reflect a consensus of the social values of its stakeholders if they are to be useful in guiding the business toward sustainability.

Non-traditional management strategies, such as those outlined in *Beloved Economies* [47], may be uncomfortable at first for some SBMs, but personal connections with stakeholders are essential in developing the metrics for managing socially sustainable businesses. Businesses in the beloved economy share decision-making, prioritize relationships, respect history, seek differences, value multiple ways of knowing, take time to make good decisions, and prototype their ideas early and often. Economic viability is essential for sustainability but the economic decisions of SBMs must be tempered by a commitment to shared leadership, shared responsibilities, and shared rewards that reflect the shared social and moral values of stakeholders.

Social values are derived from moral values. Questions of right or wrong and good or bad in social relationships are ultimately rooted in ethics, meaning individual and collective assessments of morality. The basic principles of moral right relationships among people include honesty, fairness, responsibility, compassion, and respect. These principles have been valued by people of a wide variety of religions, ethnicities, nationalities, ages, and economic and social status in many different parts of the world [48]. People who are dishonest, unfair, irresponsible, uncaring, and disrespectful cannot expect to sustain positive relationships with other people. Businesses that are unfair, dishonest, unfair, irresponsible, uncaring, and disrespectful with their employees, suppliers, customers, or investors are not morally sustainable, no matter how economically successful they may appear in the short run. SBMs need to find socially acceptable ways to carry out periodic morality checks with their stakeholders to monitor the moral sustainability of their businesses.

Sustainable relationships with nature are rooted in an ethic of sustainability. The ethic of sustainability is anthropocentric in that it relates particularly to the wellbeing of humanity, but it is also ecocentric in that it recognizes that the wellbeing of humanity is integrally interconnected with the wellbeing of the other living and nonliving things of the Earth. Principles of right relationships with nature are not as well developed as those for relations among people but can be derived from right relationships among humans. Ecologist Aldo Leopold's "land ethic", for example, "expands the definition of community to include not only humans, but all of the other parts of the Earth, as well: soils, waters, plants, and animals [49]". The principles of sustainable relationships with nature can be derived by extending the principles of right human relationships across generations.

One of the most fundamental moral principles is the Golden Rule: To treat others as you would want to be treated if you were them and they were you. The Golden Rule "dates back to the early Confucian times and can be found in most of the world's biggest religions and just about every ethical tradition" [50]. Sustainable business managers must be willing to treat their workers, suppliers, customers, community members, and other stakeholders as they would want to be treated if they were stakeholders rather than managers.

The ethic of sustainability extends the Golden Rule across as well as within generations. Those who manage and work for SMBs should use the physical and biological resources of nature and protect the natural environment as they would want the resources of nature to be used if they were members of some future generation and those of future generations were managing and working for their businesses today. Business decisions that reflect the multigenerational ethic of sustainability may result in positive economic and social outcomes, but they must be motivated by a moral sense of responsibility to help meet the needs of all in the present while leaving equal or better opportunities for those of the future.

## 9. Conclusions

Economic decisions have social and moral consequences, regardless of the motivation. Everything of economic value is ultimately derived from nature by way of society and thus economic decisions affect society and nature. Even the economic value created by human thinking and creativity depends on society and nature. Humans are biological and social beings that acquire much of their knowledge from society and their energy from other living things of nature. The human brain claims about 20% of the energy used by the human body [51].

The hierarchal relationships among nature, society, and economy are shown in Figure 3. Humans are but one among many species that occupy the Earth, and everything that sustains human life ultimately comes from the Earth—the air, water, minerals, soil, and the biological ecosystems and organisms of the Earth. Human societies are wholly contained or nested within nature, as depicted in Figure 3. The most basic principles of social relationships are principles of human nature.

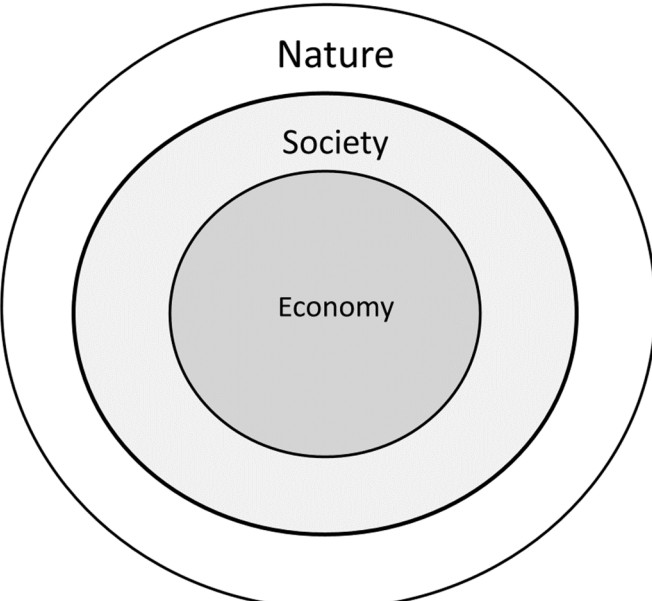

**Figure 3.** Hierarchy of Sustainability.

Economies are wholly contained or nested within societies, which in turn are nested within nature. Societies create economies to meet needs that can be met through impersonal transactions rather than personal relationships or acts of altruism. The most basic principles of economics also are principles of individual human nature. Businesses are nested within economies, which are nested within societies, which are within nature. The sustainability of businesses ultimately depends on the sustainability of the economies, societies, and natural ecosystems of which they are a part.

Every business management decision has impacts, positive and/or negative, on the societies and natural ecosystems within which they function. Even businesses that are managed for the economic bottom line create social and ecological benefits when

they provide employment and make efficient use of scarce natural resources. However, businesses that are managed to maximize profits or shareholder value should not be expected to reflect the social and moral values of sustainability in their decisions. Their only sense of social responsibility is "to use its resources and engage in activities designed to increase its profits so long as it stays within the rules of the game, which is to say, engages in open and free competition without deception or fraud [13]". Managers of large, publicly traded corporations are legally obligated to serve the "common interest" of their investors or shareholders. The shareholders of such corporations may be ethically and socially responsible people, but they have a wide diversity of social and ethical values. The only value they have in common is their desire to maximize their economic return on investment.

In addition, many investments in large corporations are made through mutual funds, and mutual funds and the individual investors may not know how many of which company's shares they own at any point in time. With computer trading, shares may be bought and sold within minutes or seconds. So, in today's corporate environment, there is no way for managers of large corporations to know or reflect the social or moral values of their investors in their decisions. Corporate managers are legally obligated to serve the common interest of their shareholders. For large, publicly traded corporations, this means maximizing profits and returns on investment.

Those who manage, work for, or are affected by economically motivated corporations must rely on the government to minimize the negative social and ecological impacts of their corporations by "setting rules of the game" and ensuring that corporations "engage in open and free competition without deception or fraud"—using Friedman's concept of corporate social responsibility [13]. Increasing economic and social inequities and existential threats such as climate change are convincing some prominent free-market economists to conclude that the current rules of the game are inadequate [52]. If all corporations were required to abide by the same rules and to engage in free and open competition, there would be no competitive disadvantage associated with rules that reflect social and ecological responsibility. However, there is no economic incentive for businesses to help establish such rules and it is to their short-run economic advantage to resist them.

The fundamental difference between businesses that are managed for the economic bottom line business is that decisions of businesses managed for sustainability, SMBs, are motivated by social and moral values as well as economic values. Furthermore, moral values are given priority over social values, and social values are given priority over economic values, as depicted in Figure 2. All three are essential for sustainability, but economic viability provides a means of creating positive economic and social benefits for workers, suppliers, customers, investors, communities, and societies. Societies provide a means for people to express a common commitment to care for each other and to care for the Earth for the benefit of those of the present as well as future generations. Sustainability is ultimately a matter of morality.

Businesses that are managed for sustainability must produce things of economic value but cannot be expected to compete economically with businesses that are allowed to extract and exploit to maximize short-run profitability. Situations may exist where SMBs can increase profits by using resources more efficiently, substituting renewable for nonrenewable resources, increasing worker productivity, and meeting consumer demand for sustainably produced products [39]. However, such situations are likely to be short-lived as profit-maximizing corporations find ways to exploit the same economic opportunities by creating the illusion of sustainability without paying the full economic costs of socially and ecologically responsible production. The industrialization of organic food production in the United States provides a prime example [53]. Today's threats to sustainability are the logical consequences of economics and societies that prioritize economics over society and society over nature: economic value over social value and social value over moral value. It is cheaper economically, meaning in the short run, to extract and exploit than to conserve and care for nature and society.

Sustainable businesses will be more profitable in the long run because unsustainable businesses are not economically viable and will become less competitive as their resources are degraded and depleted. However, profitability is determined by economic value, and the economy prioritizes the present over the future. Consequently, customers of sustainable businesses must be willing and able to pay more money for their products, workers must accept less money for their work, suppliers pay more for natural resources and accept smaller profit margins, managers accept lower salaries relative to their workers, and investors accept lower returns on their investments. In some situations, the differences in costs, prices, wages, salaries, and returns on investment will be near those of profit-maximizing corporations, but in others, the differences will be significant. Stakeholders of SMBs must realize and accept that any sacrifice in economic returns will be more than offset by the social and moral benefits that accrue to the stakeholders in sustainably managed businesses.

SBMs with shares traded in stock markets are particularly vulnerable to being taken over by economically motivated investors. Managers of family-owned corporations or closely held corporations can reflect the ethical and social priorities of their investors in their management decisions. However, SFBs that are traded on the stock exchanges may be identified as "underperforming companies" and become the subject of buyouts or hostile takeovers by investors who view them as opportunities to maximize short-run profits and inflate their stock values. Being organized as a Benefit Corporation may provide some protection, but a preferable solution might be to establish a separate stock exchange in which sustainably managed businesses could be capitalized by investors who share the social and ethical values of sustainability.

Regardless of their answers, those who manage businesses for sustainability must continually ask the question of themselves and their stakeholders: How can we best meet the needs of all in the present while ensuring equal or better opportunities for those of the future? Their collective answers to this question will be the best guide to making business decisions for sustainability.

**Funding:** This research received no external funding.

**Acknowledgments:** Many of the concepts in this paper evolved from discussions with professional colleagues Lonnie Gamble of Maharishi International University and Travis Cox of Naropa University.

**Conflicts of Interest:** The author declares no conflict of interest.

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
