# Peer review of "Business Management for Sustainability"

_sustainability, doi:10.3390/su16093714_

Round 1
Reviewer 1 Report
Comments and Suggestions for Authors The study entitled "Business Management for Sustainability" addresses an interesting and topical research topic. The author has oriented attention to the problem of sustainable management with a system of environmental and social responsibility metrics, drawing attention to the system of fundamental differences between economic, social, moral or ethical values.However, the layout of the study needs to be rethought in terms of strengthening the scientific side. I make specific comments below.
1. The article does not expose a clear research question. In this respect, the article should be improved.
The abstract should expose the purpose of the study, in response to an identified gap, which is not explicitly stated.
2.The content of the article explains the evolution and importance of governance for sustainable development in an original way, but the research gap is not clearly identified.
The introduction should set out the background of the paper, address the purpose and the gap more broadly, which should be discussed according to the nature of the paper. 3. The study is interesting because it provides a broad overview of the evolution of the sustainable management approach compared to other published works, while the theoretical nature of the work requires a very important reference to the literature and is supported by references. It is worth discussing contemporary business models in relation to the economic background - particularly when the text discusses the specific historical background of sustainable management. In this respect, it is worthwhile to study https://doi.org/10.3390/su15118889, https://doi.org/10.3390/su141811695, among others. It is worthwhile to review the main indicators in each area in order to strengthen the study from a research point of view, as an important part of the study discusses their theoretical side. 4. In addition to strengthening the theoretical side, the main improvement of the study must be the strengthening of the research side. It lacks a prominent research thread (assumption). There is a lack of discussed research methodology - even if the reference basis is a literature review, this is necessary.
5. The paper lacks an exposed discussion section in terms of the findings developed with existing research results in the topic explored. As already mentioned, the theoretical section is insufficiently referenced to the literature (references).
6. The literature should be significantly strengthened (number of references). Additional comments:
Make the summary section shorter and oriented to demonstrate the contribution of the study to theory. The study is interesting, extensive on the theoretical side. The highlight page of the scientific text needs strengthening.
Author Response
Response:
- The abstract has been completely rewritten to explain the purpose of the paper and identify the conceptual gap that is addressed in the paper. There is no research question to be exposed because the paper is a perspective paper rather than a research paper.
- The paper is a perspective paper rather than a research paper and thus there is no research gap to be identified.
- The paper is not a review paper and thus does not require an extensive review of the literature. The overview of the evolution of corporate responsibility was intended to provide a conceptual foundation for the rest of the paper rather than document the current state of knowledge concerning sustainable management strategies. The two suggested references have been included in a new introduction section on Page 2 to clearly identify that the purpose of the paper is to present a perspective rather than provide a research report or review of literature.
- There is no research thread or methodology in the paper because it is not intended to be a research paper, but instead a perspective on an important aspect of sustainability. The complexities of matters related to sustainability are not compatible with generally accepted research methodology, which is addressed on page 16 in the Implications section.
- Again, the paper is not intended to be a research report or review of literature but the presentation of a perspective on an important issue of sustainability.
- The Implications and Conclusions sections focus on the importance of differences in economic, social, and moral values in managing sustainable businesses and the essential role of society and government in creating an environment in which businesses can be managed sustainably. These are the primary contributions of the paper. Perspectives of sustainability cannot be proven or disproven by scientific studies or reviews of literature but must be evaluated and accepted or rejected by relying on logic and rationality.
Reviewer 2 Report
Comments and Suggestions for Authors
The paper provides a coherent analysis of how moral or ethical values, social values and economic (profit-maximization) values for shaping today's Sustainable Business Management. The paper provides a clear focus on the interplay of these three level of values where moral values are placed on the top the pyramid. The arguments and implication for the design and implementation of sustainable business strategies. I suggest that it would be nice to add 1-2 paragraphs on investors' perceptions on the management strategies that consider not only economic values but also social and moral values by today's firms.
Author Response
These comments have been very helpful in rewriting the abstract of the paper to focus more clearly on the most important aspects of the paper. New Figures have been added on pages 12 and 18 to illustrate the hierarchies of moral, social, and economic values and of nature, society, and economy.
As suggested, two paragraphs have been added, on pages 3 and 4, to address investors' perspectives on today's sustainably managed corporations. The perspectives of consumers and workers are now included as well. These paragraphs, along with other revisions, significantly strengthen the discussion of impact investing and socially responsible investing. However, the discussion is still intended to be a brief overview rather than a review of literature as is now clearly explained in the revised paper.
Reviewer 3 Report
Comments and Suggestions for Authors
The paper proposes a broad-spectrum overview of the topic of managing businesses for sustainability. After introducing the notion of corporate responsibility, the paper presents key concepts related to sustainable business management along five different areas: (i) business metrics; (ii) motives; (iii) economies; (iv) economic strategies; (v) implications. The paper is dense in content and it is evident that the author has significant knowledge of the field of sustainability. The illustration of the presented concepts is quite generic, but that is probably inevitable given the broad-spectrum perspective adopted for the paper. As a reviewer, I have been asked to provide some suggestions to improve the paper. Hence, below I offer some ideas that, in my opinion, would improve the appeal and readability of the paper. I hope that the authors will find them useful.
1. I think it is necessary to define the objective of the paper. It is currently unclear how the paper contributes to the larger scheme of providing insights into the topic of business management for sustainability. In my opinion, the purpose and contribution of the paper should be stated in the abstract and illustrated in greater detail in the introduction. What should we expect to learn from reading this paper? What is the novel contribution that the paper provides? Where do its uniqueness and novelty lie? I think it would be important to provide a clear and explicit answer to these questions.
2. I think it would be useful to briefly present the structure of the paper (perhaps at the end of the introduction). In my opinion, the reader needs to know from the beginning that the paper will focus on those five areas – and, ideally, also the reason why the author identified or chose those specific areas.
3. I think it would be useful to organize the sections into sub-sections. As the paper stands now, the logical flow is difficult to follow, and the reader gets lost very easily. For example, section 5 “Three economies of sustainability” is 5 ½ pages long. I had to pencil notes on it in order to be able to follow the structure – at the first round of reading, it looked like only a very long list of definitions. Creating sub-sections would certainly improve clarity and help the reader point (or go back) to the concepts that are more relevant to her/his specific interests.
4. I think it would be useful, if possible, to add some figures (or tables) to help visualize the logic underlying the discussion – and hence relationships, hierarchies, influences etc. among the presented concepts.
5. Some paragraphs seem to be misplaced. This is the case in particular of paragraphs starting with “Investment in businesses committed…” on page 2 (lines 66-71) and “In managing businesses for sustainability…” on page 6 (lines 242-249). The notions of IIs/SRIs and value could probably be more effectively introduced close to where they are first used in the paper. I have the same problem with the paragraphs between lines 105 and 112 on page 3. Moreover, I struggle to see how TPM (lines 113-120) fits with the proposed discussion of corporate responsibility.
6. I have noted some minor errors throughout the paper. On page 4 (line 153), it should be “Social Responsibilities of the Businessman” rather than “Socially Responsibilities of the Businessman”. On page 4 (line 162-163), the word “investment” seems redundant. On page 14 (line 591) the citation [42] should be placed right after “Beloved Economies” rather than at the end of the sentence.
Author Response
These comments and suggestions have been particularly helpful in revising the paper. The purpose of the paper and how the various sections fit together to address the purpose are now clearly identified in a rewritten abstract and a new introduction to the paper.
- The abstract now identifies the purpose of the paper to let readers know what they can expect to learn from the paper, the novelty of the paper in explaining the differences between economic, social, and moral values, and the critical importance of these differences in managing sustainable businesses. Neither an acknowledgment of the differences in value nor the appreciation of the importance of these differences is apparent in currently prominent approaches to managing sustainable businesses.
- A new Introduction to the paper has been added in which different sections of the paper are identified and how they are intended to fit together to fulfill the purpose of the paper is explained.
- Three subsections have been added to break up the long section on The Three Economies of Sustainability, the first addressing motives, the second values, and the third morality and sustainability. This is the longest because it contains the unique or novel contributions of this paper and is likely to be the most difficult to fully comprehend or appreciate.
- New figures have been added on pages 12 and 18 to illustrate the hierarchal relationships among moral, social, and economic values and nature, society, and economy.
- Several paragraphs have been moved to improve the flow of the paper. The paragraph on Benefit Corporations has been moved above a rewritten and expanded discussion of impact investing and socially responsible investing on pages 3 and 4. Taylor's contributions to corporate responsibility are explained more fully on page 3. The previous introductory paragraph in the Three Economies section has been moved to the beginning of the Values subsection. These and other significant changes are highlighted in the revised.
- The typographical errors noted have been corrected and the revised paper has been proofread in an attempt to correct any additional typographical and grammatical errors. The paper still may not be perfect but a serious effort has been made to respond to this concern.
Round 2
Reviewer 1 Report
Comments and Suggestions for Authors
The article has been improved